# *Acanthamoeba* keratitis cases in Sweden: genotypes and clinical course

Elin Karlsson,[1,2] Leigh Davidsson,[3] Per Montan,[1,2] Emma Nivenius[1,2]

**ABSTRACT** *Acanthamoeba* keratitis is a sight-threatening corneal infection. A delay in diagnosis and lack of amoeba-specific treatment contribute to a generally poor prognosis. Genotyping of pathogenic *Acanthamoeba* may offer deeper insights into the disease. Sixteen culture-positive *Acanthamoeba* ocular samples from 2006 to 2022 were identified and genotyped in patients from the northern and central regions of Sweden. Data from the patients' records were collected, and their respective clinical courses were reviewed. The most common *Acanthamoeba* genotype was T4 (11 cases), followed by T3 (4 cases) and T6 (1 case). All but one patient was a contact lens user. There were no cases of previous trauma. No statistically significant differences were found between the T4 and non-T4 groups. The influence of genotype on the clinical course warrants further investigation.

**IMPORTANCE** This study provides insights into the rare but serious eye infection *Acanthamoeba* keratitis, which can cause blindness. Our research presents data from Sweden on the genotypes of the parasite causing this infection. While the most common genotype worldwide, T4, was also predominant in our study, we found an unexpectedly high number of cases caused by the T3 genotype. This may indicate regional differences and highlights the need for ongoing research. The study also examines the link between genotype and patient outcomes. Although our findings did not establish a statistically significant correlation, knowledge of such a connection could prove valuable in efforts to improve management, and further multicenter studies may be warranted.

**KEYWORDS** *Acanthamoeba*, keratitis, genotypes

*A*canthamoeba is a free-living protozoan that is widespread in the environment globally, being found in soil, water, and dust. The amoeba alternates between two stages: an active, mobile trophozoite form and a highly resistant cyst form that develops in harsh environments. *Acanthamoeba* can cause a severe sight-threatening corneal infection in otherwise healthy individuals and a fatal brain and spinal cord infection in immunosuppressed patients.

*Acanthamoeba* keratitis (AK) is rare. Its prevalence varies across the world. In developed countries, contact lens wear is the main risk factor (1, 2), while in developing countries, trauma is the most common risk factor (3, 4). A review from 2023 estimated the global annual incidence of AK to be 2.9 cases per million people (5). There have been reports of an increasing incidence in Western countries (6–10). This rise is primarily thought to be linked to the growing use of contact lenses. In addition, heightened suspicion and enhanced diagnostics may contribute to the increase in the number of identified cases.

There are several types of *Acanthamoeba*. Traditionally, the genus *Acanthamoeba* has been classified into species. At least 30 species have been defined, and these are further divided into three groups (I–III) according to morphological characteristics, such

**Peer Reviewer** Christopher Aaron Rice, Purdue University, West Lafayette, Indiana, USA

Address correspondence to Elin Karlsson, elin.karlsson@ki.se.

The authors declare no conflict of interest.

See the funding table on p. 8.

as the shape and size of the cysts, as described by Wang et al. (11). This classification of *Acanthamoeba* has limitations because the morphology of the amoeba changes in different stages throughout its life cycle. A new classification based on regions of the 18S rRNA has been developed with the advancement of sequencing. The current classification includes 23 genotypes or sequence types labeled from T1 to T23. Some of them, such as T4, are further divided into subgroups (12). Studies on *Acanthamoeba* in the environment from various regions of the world have proven T4 to be the most widespread genotype (13–16). Additionally, T4 is recognized as the most common genotype in cases with AK (5, 17) and in other *Acanthamoeba* infections (18). Whereas several genotypes are known to be pathogenic (T1 to T13, T15, T16, and T18), others have not yet been identified in diseases (T14, T17, and T19 to T23) (11). Differences in virulence factors between pathogenic and non-pathogenic *Acanthamoeba* strains have been demonstrated, but differences between pathogenic genotypes have not been shown (19).

Despite the growing number of publications on AK genotypes, few have explored the relationship between genotype and clinical outcome (20–26). In this study, we aim to investigate the genotypes of *Acanthamoeba* keratitis cases in Sweden and to assess any relationship between genotype and the clinical course.

## MATERIALS AND METHODS

The Public Health Agency in Solna, Stockholm, analyzes all samples from suspected AK cases from the northern and central regions of Sweden. This study examined all isolates from culture-positive *Acanthamoeba* keratitis cases identified from 2006 to 2022.

### *Acanthamoeba* culture

At the time of keratitis, clinical samples were inoculated onto non-nutrient agar plates seeded with *Escherichia coli*. Agar plates were incubated at 30°C for 1 week. Cultures were investigated microscopically for the presence of trophozoites and cysts of *Acanthamoeba* species with characteristic morphology. Positive cultures were subsequently stored at 4°C.

### DNA extraction

Prior to extraction of parasite DNA, the cysts were mechanically broken for 1 min in a Bullet Blender at maximum speed. Fifty microliters of cyst solution was mixed with 500 µL of buffer G2 (Qiagen, Hilden, Germany), 50 µL of proteinase K solution (Qiagen, Gdansk, Poland), and a volume that equated to 250 µL of Lysing matrix D (MP Biomedicals, Ohio, USA). After mechanical disruption, the solution was centrifuged at 6,000 × *g* for 10 min. The supernatant (400 µL) was incubated in a shaking heating block at 56°C for 1 h. The full volume was used for DNA extraction in a PSS MagLead 12gC robot (Bioservices, Stockholm, Sweden).

### Sequence typing

The *Acanthamoeba*-specific primer pair initially used for sequence typing used the forward primer JDP1 (5′-GGCCCAGATCGTTTACCGTGAA) and the reverse primer JDP2 (5′-TCTCACAAGCTGCTAGGGAGTCA) (27). Depending on the genotype, the primers amplified a product of ca. 423–551 bp of 18S rDNA. The amplification was performed in a volume of 50 µL containing a mixture of 1× Qiagen PCR buffer with 1.5 mM MgCl$_2$ (Qiagen), extra 1 mM MgCl$_2$, 0.2 µM primers, 0.2 mM dNTP, 10 µL Qiagen Q solution, and 1.5 U HotStarTaq Plus DNA polymerase per reaction tube. Five microliters of DNA template was used in the PCR reaction. The reaction mixes were incubated for 3 min at 94°C, followed by 45 cycles of 30 s at 94°C, 1 min at 61°C, and 1 min at 72°C. A final cycle for 10 min at 72°C was included. The Veriti 96-Well PCR machine was used for PCR. All samples were initially sequenced using this PCR product. For samples with

suboptimal sequencing quality, additional PCR reactions were performed to enable reliable sequence typing. A PCR targeting the 1,375 bp region of the 18S rDNA was conducted. In this second PCR, the eukaryote-specific primer pair used the forward primer CRN5 (5′-CTGGTTGATCCTGCCAGTAG) and the reverse primer 1137 (5′-GTGCCCTT CCGTCAAT), which amplified a product of ca. 1,472 bp of the 18S rDNA region (27). The amplification was performed in a volume of 50 µL containing a mixture of 1× Qiagen PCR buffer with 1.5 mM $MgCl_2$ (Qiagen), extra 1 mM $MgCl_2$, 0.2 µM primers, 0.2 mM dNTP, 10 µL Qiagen Q solution, and 1.5 U HotStarTaq Plus DNA polymerase per reaction tube. Five microliters of DNA template was used in the PCR reaction. The reaction mixes were incubated for 7 min at 95℃, followed by 45 cycles of 1 min at 95℃, 1 min at 60℃, and 2 min at 72℃. A final cycle for 10 min at 72℃ was included. A third PCR reaction was also used for sequence typing of these samples. The primer pair used the forward primer AcanSeqF (5′-CCTACCATGGTCGTAACGGG) and reverse primer AcanSeqR (5′-AGGG CAGGGACGTAATCAAC), which amplified a product of ca. 1,750 bp of the 18S rDNA region (28). The amplification was performed in the same manner as the eukaryote-specific PCR method above.

The following primers were used for single direction Sanger sequencing using Big Dye Terminator chemistry: 892C (5′-GTCAGAGGTGAAATTCTTGG), 373 (5′-TCAGGCTCCC TCTCCGGAATC), 570C (5′-GTAATTCCAGCTCCAATAGC), and JDP1 and JDP2. Genotyping was attempted on 21 samples and succeeded in 17 samples; of these, 16 were eligible for this study.

## Phylogenetic analysis

Sequences from the Swedish isolates were compiled and analyzed in CLC Main Workbench (version 23.0.02). The 18S rDNA sequences were subject to an NCBI BLAST search (https://blast.ncbi.nlm.nih.gov/Blast.cgi) in order to determine the sequences with the highest homology to the Swedish isolates. In addition, a master alignment of 18S rDNA sequences was created using 54 *Acanthamoeba* reference sequences, including genotypes T1–T23 (see Addendum, Table S1). Isolate and reference sequences were aligned with MEGA7 (version 7.0.26), and sequence analysis was performed in CLC Genomics Workbench (version 21.0.4). Neighbor-joining distance trees were obtained using MEGA, and trees were rooted with the *Acanthamoeba comandoni* ATCC30135 (AF019066) sequence.

## GenBank accession numbers

The sequences of the Swedish isolates are designated by the following accession numbers in GenBank: PQ002557–PQ002573.

## Review of clinical course

The patients' medical records were obtained and reviewed. A study chart including background data, putative risk factors, disease-specific factors, treatment, and clinical outcome was created. Best corrected visual acuity (BCVA) at the final visit was denoted in Snellen decimal and converted to logarithm of the minimum angle of resolution (LogMAR). The conversion to LogMAR for counting fingers, hand movements, perception of light or amaurosis was performed by the method established by Day et al. (29)

## Statistical analysis

Variable distributions in the T4 and non-T4 cohorts were analyzed using non-parametric tests for numerical and categorical data (Statistica, version 14.1.08), with results shown only for one difference.

## RESULTS

### Culture and genotyping

From 2006 to 2022, 26 corneal samples were positive for *Acanthamoeba* species by culture at the Public Health Agency of Sweden. Twenty-one samples were available for this study, and DNA was successfully extracted and sequenced in 17 of them. One case was excluded from the study because the *Acanthamoeba*-positive sample was obtained from contact lens solution, while the corneal scrape grew *Pseudomonas aeruginosa*, and the keratitis resolved promptly with antibiotics alone (isolate PHAS 11-2 KLAR in Fig. 1).

The sequence analysis results with the GenBank BLAST nucleotide database are presented in Table S2 Addendum. The results of neighbor-joining distance trees, showing the Swedish isolates clustering with the reference sequence types, are illustrated in Fig. 1 and 2. The most common *Acanthamoeba* sequence type infecting Swedish patients was T4 (11/16 clinical isolates, 69%). Five samples were of non-T4 sequence type: four isolates (25%) clustered with the T3 reference strains, and one isolate was identified as sequence type T6 (6%). The T4 strain was subclassified (see Addendum, Table S2).

### Clinical results

The study presents 16 cases of culture-confirmed *Acanthamoeba* keratitis (see Table 1). Bilateral AK was suspected in one case, but only one eye was sampled (T4) and included in this study. The remaining cases were unilateral (94%). The male:female ratio was 3:5, with a median age of 30.5 years (range 18–56 years). One case lacked information regarding contact lens usage; however, the remaining 94% were confirmed contact lens users.

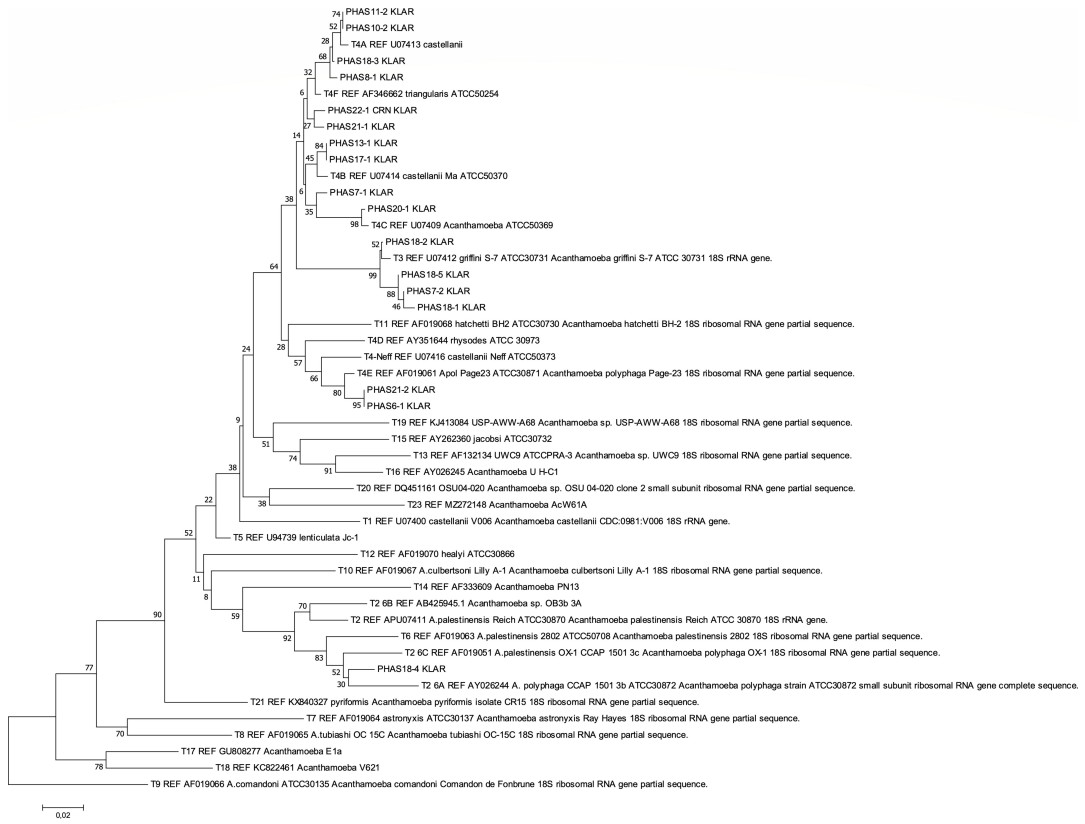

**FIG 1** Neighbor-joining tree showing *Acanthamoeba* isolates from Swedish patients clustering with reference sequence types based on a 257 bp region of the 18S rDNA gene. Branch lengths are proportional to genetic distance; the scale bar represents a distance of 0.02.

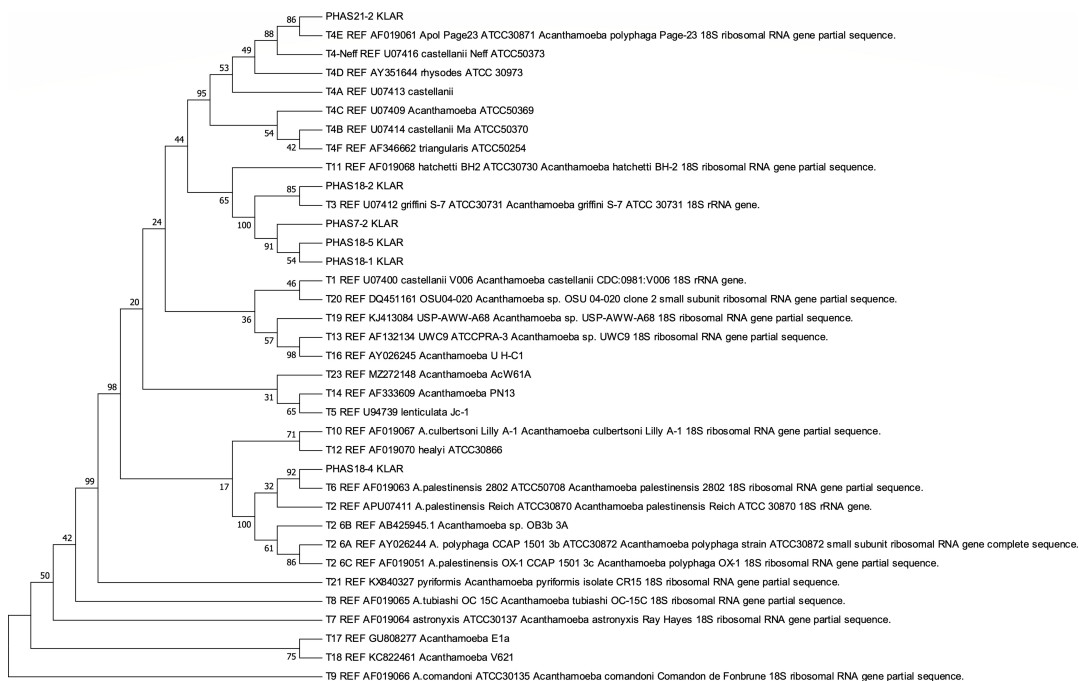

**FIG 2** Neighbor-joining tree showing *Acanthamoeba* isolates from Swedish patients clustering with reference sequence types based on a 1,375 bp region of the 18S rDNA gene.

Cases were distributed across the entire geographical study area in Sweden (see Fig. 3). Previous eye disease was found only in three patients (19%). In four cases, a history of swimming shortly before the onset of ocular symptoms was reported.

AK was not suspected at the first visit in any of the cases. The most frequent initial diagnosis was viral keratitis (75%), and, in most cases, two or more other diagnoses were considered before the suspicion of AK was raised. Time to correct diagnosis from onset of symptoms was within 1 month for 45% of the T4 group and for 40% of the non-T4 group. At the time of diagnosis, three of the T4 cases showed only epithelial changes, and the remaining eight had stromal engagement to some extent. In the non-T4 group, all patients had stromal affection with complete or incomplete ring infiltrate at the time of sampling.

All patients received a combination of available antiamoebic antiseptic drops, the most common combination being chlorhexidine (0.02%) and polyhexamethylene biguanide (0.02%). For the T4 group, the median treatment time was 8 months (range: 2–14 months), and that for the non-T4 group was 11.25 months (range: 7.5–35 months). Surgical treatments, such as amniotic membrane transplantation, corneal cross-linking, and penetrating keratoplasty, were opted for in half of the patients in both groups.

At the last visit, the BCVA was worse than 1.0 LogMAR in 44% of the patients. Among the non-T4 cases, 80% had a BCVA worse than 1.0 LogMAR, compared to 27% of the T4 cases. This difference was not statistically significant (Mann-Whitney U-test, $P = 0.3$).

## DISCUSSION

This retrospective study presents 16 culture-proven AK cases, providing information about the clinical course and prevalence of different genotypes in Sweden. Contact lens wear was the predominant risk factor. The genotyped cases exhibited a uniform distribution of T4 and non-T4 cases throughout the study area, with a greater incidence observed in densely populated regions.

In a systematic review from 2021, Diehl et al. summarized the findings from genotyping of AK from 2002 to 2020, representing 27 countries worldwide. T4 was found to be the most prevalent genotype on all continents (17). The results of our study align with

**TABLE 1** Characteristic features of the *Acanthamoeba* keratitis patients including genotype data[b]

| Patient | Genotype | Sex | Age (years) | Contact lens | Severe pain | Initial diagnosis | Topical steroids before diagnosis | Time to diagnosis[a] (month) | Appearance of the keratitis at time of AK-sampling | Surgery | BCVA at the last visit (Snellen decimal) | LogMAR at the last visit |
|---|---|---|---|---|---|---|---|---|---|---|---|---|
| 1 | T4 | M | 46 | Yes | No | Herpes | Yes | 1–3 | Stromal | None | 0.3 | 0.5 |
| 2 | T4 | F | 23 | Yes | Yes | Undefined virus | Yes | 1–3 | Stromal | CXL | 0.1 | 1 |
| 3 | T4 | M | 56 | Yes | Yes | Herpes | Yes | 1–3 | Stromal | CXL, amniotic MT, enucleated | A | 3 |
| 4 | T4 | M | 41 | Yes | No | Scar after foreign body | Yes | 1–3 | Epithelial | None | 0.6 | 0.2 |
| 5 | T4 | F | 47 | Yes | Yes | Herpes | No | <1 | Stromal | None | 1.0 | 0 |
| 6 | T4 | F | 59 | Yes | Yes | Inflammatory | Yes | <1 | Stromal | Eviscerated | A | 3 |
| 7 | T4 | M | 34 | Yes | Yes | Herpes | No | <1 | Epithelial | None | 1.0 | 0 |
| 8 | T4 | F | 25 | No information | Yes | Herpes | NA | >3 | Stromal | CXL, amniotic MT, penetrating KP | 0.4 | 0.4 |
| 9 | T4 | F | 24 | Yes | No | Virus | Yes | >3 | Stromal | CXL | HM | 2.4 |
| 10 | T4 | F | 24 | Yes | Yes | Herpes | Yes | <1 | Epithelial | None | 1.0 | 0 |
| 11 | T4 | F | 18 | Yes | Yes | Inflammatory | Yes | <1 | Stromal | None | 0.16 | 0.8 |
| 12 | T6 | F | 21 | Yes | Yes | Herpes | Yes | >3 | Stromal | Amniotic MT | 0.02 | 1.7 |
| 13 | T3 | M | 27 | Yes | Yes | Herpes | No | <1 | Stromal | None | CF | 2.1 |
| 14 | T3 | F | 55 | Yes | Yes | Inflammatory | Yes | 1–3 | Stromal | Amniotic MT | HM | 2.4 |
| 15 | T3 | M | 46 | Yes | Yes | Undefined virus | No | <1 | Stromal | None | 0.9 | 0.04 |
| 16 | T3 | F | 26 | Yes | Yes | Virus | Yes | 1–3 | Stromal | Amniotic MT | LP | 2.7 |

[a]From onset of symptoms.
[b]A, amarousis; CF, counting fingers; CXL, corneal cross-linking; HM, hand movement; KP, keratoplasty; LP, light perception; MT, membrane transplant; NA, not available.

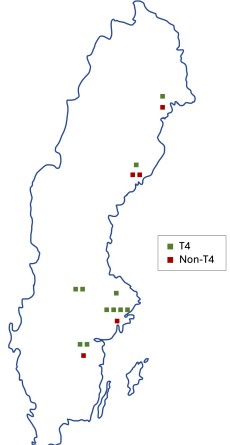

**FIG 3** Geographical distribution of 16 *Acanthamoeba* keratitis cases from central and northern regions of Sweden.

this review, showing that T4 is the most common genotype in Swedish cases, followed by T3 as the second most common genotype.

The T3 genotype appears to be more frequent in Europe than in other parts of the world. Given its geographical distribution, it could be speculated that wearing contact lenses, rather than exposure to corneal trauma, might predispose individuals to T3 genotype infection. In our study, four out of the five non-T4 cases were of the T3 genotype, and all non-T4 cases were contact lens wearers. A recent publication from our neighboring country, Denmark, reports 38 cases of AK, of which 35 were confirmed as T4 and 3 were caused by T3 (26). Interestingly, in three Asian studies, trauma was identified as the primary risk factor, and no cases of the T3 genotype were identified (21, 22, 25).

Although there are numerous research and case reports on the genotypes of AK cases, only a few have presented study populations with clinical characteristics and outcomes that allow for some interpretation of the relationship between genotype and clinical course (20–26). The distributions of genotypes in these studies are presented in Table 2. In fact, only three of the studies (22, 23, 26) include more than one single non-T4 case in the study cohorts. In one of the articles, it was tentatively proposed that the non-T4 genotype had a worse clinical outcome (23).

**TABLE 2** Studies on *Acanthamoeba* keratitis genotype and clinical course

| Author, year of publication | Country of origin | Study years | Numbers of patients | Genotypes | Genotype in relation to outcome |
|---|---|---|---|---|---|
| Zhao et al., 2010 (25) | China | 2000–2009 | 14 | T4: 14 cases | Analysis not possible |
| Arnalich-Montiel et al., 2014 (23) | Spain | 2009–2013 | 17 | T4: 14 cases | Bad outcome in non-T4 cases |
| | | | | T3: 2 cases | |
| | | | | T11: 1 case | |
| Orosz et al., 2019 (20) | Hungary | 2015–2018 | 7 | T4: 6 cases | Analysis not possible |
| | | | | T8: 1 case | |
| Roshni et al., 2020 (22) | India | 2016–2018 | 30 | T4: 26 cases | No difference between genotypes |
| | | | | T12: 3 cases | |
| | | | | T11: 1 case | |
| Rayamajhee et al., 2022 (21) | India | 2020 Feb - Oct | 13 | T4: 12 cases | Analysis not possible |
| | | | | T12: 1 case | |
| Pang et al., 2024 (24) | China | 2014–2023 | 159 | All T4, different subtypes studied | Analysis not possible |
| Skovdal et al., 2026 (26) | Denmark | 2002–2022 | 38 | T4: 35 cases | No difference between genotypes |
| | | | | T3: 3 cases | |

In summary, the number of published non-T4 cases is small, and the effect of genotype on the clinical course is unclear. It should also be acknowledged that factors other than specific genotypes are important determinants of the outcome. Early diagnosis and treatment are decisive for the prognosis (30), especially if the disease is recognized and managed in its epithelial stage (31). Unfortunately, the initial clinical signs are unspecific and misinterpreted, which the present investigation attests to. AK was not the first suspected diagnosis in any of our cases.

No differences in the time from onset of symptoms to diagnosis were observed between the T4 and non-T4 cases. Regardless of genotype, a diagnosis within 1 month was associated with better final BCVA in our study. Still, the prognosis overall was poor, with 44% ending up with a BCVA less than 1.0 LogMAR and only 25% reaching a BCVA of 0.04 LogMAR or better. Despite the lack of statistical significance ($P = 0.3$), the non-T4 cases appeared to have a less favorable prognosis.

In conclusion, this study demonstrates the prevalence of different genotypes in cases of AK in Sweden. T4 is the dominating genotype, but interestingly, T3 was found in 25% (4 out of 16) of the cases. Further investigations are required to validate and specify the influence of genotype on AK pathogenicity and outcome. While *in vivo* confocal microscopy is an increasingly popular clinical support tool, we strongly emphasize that laboratory verification should always be judiciously assessed. Corneal sampling for culture or PCR is still important to confirm AK and is also of value for future genotype studies. Considering the rarity of AK in most countries, a multicenter study would be required to achieve an adequate and representative study group.

## ACKNOWLEDGMENTS

We thank Gustav Killander for help with sequence alignments and tree analysis.

The study was performed with financial support from the Swedish "Ögonfonden," the Carmen and Bertil Regnérs Foundation, and the Elsy, Harry, and Henrik Johansson Siblings Foundation.

## AUTHOR AFFILIATIONS

[1]Division of Eye and Vision, Department of Clinical Neuroscience, Karolinska Institutet, Stockholm, Sweden
[2]S:t Erik's Eye Hospital, Stockholm, Sweden
[3]Public Health Agency of Sweden, Solna, Sweden

## AUTHOR ORCIDs

Elin Karlsson http://orcid.org/0009-0007-7141-4277

## FUNDING

| Funder | Grant(s) | Author(s) |
| --- | --- | --- |
| Ögonfonden | | Per Montan |
| Carmen and Bertil Regnér's Foundation | | Elin Karlsson |
| | | Emma Nivenius |

## AUTHOR CONTRIBUTIONS

Elin Karlsson, Data curation, Investigation, Writing – original draft | Leigh Davidsson, Data curation, Formal analysis, Investigation, Methodology, Writing – original draft, Writing – review and editing | Per Montan, Data curation, Formal analysis, Investigation, Methodology, Writing – original draft, Writing – review and editing | Emma Nivenius, Supervision, Writing – original draft, Writing – review and editing

## DATA AVAILABILITY

The specimens analyzed in this study are biobanked and are available upon reasonable request through the Public Health Agency of Sweden. The accession numbers in GenBank are: PQ002557–PQ002573. The clinical and registry data collected for this study are not publicly available due to ethical and legal restrictions.

## ETHICS APPROVAL

Ethical approval for the study was obtained from the Swedish Ethical Review Authority. The research was carried out in accordance with the Helsinki Declaration.

## ADDITIONAL FILES

The following material is available online.

### Supplemental Material

**Supplemental material (Spectrum03058-25-S0001.docx).** Tables S1 and S2.

### Open Peer Review

**PEER REVIEW HISTORY (review-history.pdf).** An accounting of the reviewer comments and feedback.

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
