## [Reviewer comments · Microbiology Spectrum]

Microbiology Spectrum

Acanthamoeba keratitis cases in Sweden: genotypes and clinical course.

Elin Karlsson, Leigh Davidsson, Per Montan, and Emma Nivenius

Corresponding Author(s): Elin Karlsson, Karolinska Institute, Department of Clinical Neuroscience

Review Timeline:

Submission Date:	October 3, 2025
Editorial Decision:	December 1, 2025
Revision Received:	December 30, 2025
Editorial Decision:	January 8, 2026
Revision Received:	February 11, 2026
Accepted:	February 23, 2026

Editor: Gillian Tarr

Reviewer(s): Disclosure of reviewer identity is with reference to reviewer comments included in decision letter(s). The following individuals involved in review of your submission have agreed to reveal their identity: Christopher Aaron Rice (Reviewer #2)

Transaction Report:

DOI: <https://doi.org/10.1128/spectrum.03058-25>

Re: Spectrum03058-25 (**Acanthamoeba keratitis cases in Sweden: genotypes and clinical course**)

Dear Dr. Elin Karlsson:

Thank you for the privilege of reviewing your work. Below you will find my comments, instructions from the Spectrum editorial office, and the reviewer comments.

Please indicate in your response whether sequence data has been deposited in a public repository.

Revision Guidelines

Sincerely,
Gillian Tarr
Editor
Microbiology Spectrum

Reviewer #1 (Comments for the Author):

The manuscript reports on 16 cases of Acanthamoeba Keratitis and finds that a higher than expected number of these were of the T3 genotype, although similar results have been found before this work is solid, well written and clear.

Reviewer #2 (Comments for the Author):

Please find attached major and minor comments.

Sequence data may need deposited for others to be able to use these for constructing trees in the future.

This important article raises awareness of Acanthamoeba keratitis cases in Sweden between 2006-2022. It highlights an important observation with T4 and T3 genotypes being high on the number of AK cases seen within European countries.

Comments

Please remove the second title in line 26.

I think it would be best to include the Addendum Figures in the main text of the results of this paper.

Are these specimens biobanked? Can someone gain access to these cultured strains if they reach out? Can you start biobanking these?

At our Free-living amoeba meeting last month. There was a case of bilateral AK from two different Acanthamoeba species and genotypes. It may be best to try and get samples of both eyes in the future as we may assume it is just one, but this could complicate things significantly for the patient.

Major edits

You mention several times that 16-culture positive and then “Genotyping was attempted on 22 samples and was successful in 17 samples”. You include this in Fig 1 Addendum as sample “PHAS11-2 KLAR” but fail to include this in Table 2 or throughout the text.

It would be ideal to include the information about the isolate and update the numbers as this seems to be another T4 isolate.

It would be good to include treatment (drugs and %’s administered) and treatment timeline before the surgery column in Table 1. You may also want to include clinical outcome – resolved after Rx, resolved after surgery, enucleated, lost to follow up.

In line 232 please make a note that ~75% of cases were misdiagnosed as viral keratitis. Raising light into suspecting amoeba or other pathogens at time of first clinical presentation.

It would also be good to know if the patients were prescribed steroids before a true clinical diagnosis was made and how many this occurred to. This can exacerbate the infection and could lead to worse clinical outcomes.

Minor edits

Discrepancy between line 159 “Genotyping was attempted on 22 samples and was successful in 17 samples” and lines 192-193 “Twenty-one samples were available for this study, and DNA was successfully extracted and sequenced in 16 of them.” Where did this other sample come from? Please update records and statistics with including the missing “PHAS11-2 KLAR” sample.

Please add reference to the figures in the main text and discuss about the level of sequencing and why there are two phylogenetic trees formed.

Are you able to associate subtype and add these details into Table 2 addendum? Rather than readers trying to extrapolate this themselves from your trees.

Remove “figure E2” from bottom right of Fig 1.

Line 240 – please add concentrations of these therapies. – 0.08% of PHMB is now being adopted as the standard care based on the ODAK Phase III clinical trial in EU. SIFI will donate PHMB in EU for the treatment of AK.

Table 2 – You don’t need to have the authors written out twice. Look here; <https://journals.asm.org/journal/spectrum/article-types#research-articles>

“In the text, references are cited parenthetically by number in sequential order”. You can then use the respected number instead of writing out the authors twice. References are different font than main text.

In table 2 addendum, you could add a check box for full or partial 18S sequence and have a checkbox for which samples are then included in which figure.

Response to Reviewers

This important article raises awareness of Acanthamoeba keratitis cases in Sweden between 2006- 2022. It highlights an important observation with T4 and T3 genotypes being high on the number of AK cases seen within European countries.

Comments

Please remove the second title in line 26.

The second title on line 26 has been removed.

I think it would be best to include the Addendum Figures in the main text of the results of this paper.

The addendum figures have now been included in the main text.

Are these specimens biobanked? Can someone gain access to these cultured strains if they reach out? Can you start biobanking these?

The specimens are biobanked and it is possible to get access to them through the Public Health Agency of Sweden upon request (info@folkhalsomyndigheten.se).

At our Free-living amoeba meeting last month. There was a case of bilateral AK from two different Acanthamoeba species and genotypes. It may be best to try and get samples of both eyes in the future as we may assume it is just one, but this could complicate things significantly for the patient.

We agree that sampling from both eyes is mandatory when bilateral Acanthamoeba keratitis is suspected. In this retrospective study, however, we were limited by the information available in the medical records, and in this particular case only one eye had been sampled.

Major edits

You mention several times that 16-culture positive and then “Genotyping was attempted on 22 samples and was successful in 17 samples”. You include this in Fig 1 Addendum as sample “PHAS11-2 KLAR” but fail to include this in Table 2 or throughout the text.

Thank you for your thorough review. The correct number of culture-positive samples included in the study was 16. One sample was from contact lens solution and was excluded because it was considered a contamination; the patient was diagnosed with Pseudomonas keratitis and showed prompt improvement with antibiotics only. This is now clarified in the results section.

It would be ideal to include the information about the isolate and update the numbers as this seems to be another T4 isolate.

Information about the excluded isolate is now described more clearly in the results section. However, because this isolate was not included in the study, the number of included cases remains 16.

It would be good to include treatment (drugs and %’s administered) and treatment timeline before the surgery column in Table 1.

Thank you for this valuable suggestion. However, since multiple medication combinations were used and treatment regimens were frequently adjusted, reporting the exact sequence and proportions of the administered drugs was not feasible. Therefore, we summarised the medical treatment based on the most commonly used combinations, as described in the Clinical Results section. The concentrations of the most commonly used drops, chlorhexidine (0.02%) and polyhexamethylene biguanide (0.02%), are now specified in the text.

You may also want to include clinical outcome – resolved after Rx, resolved after surgery, enucleated, lost to follow up.

Thank you for the suggestion. The requested clinical outcome information is found in Table 1, which details the type of surgery performed for each patient, including those who underwent evisceration or enucleation. Clinical outcome is also shown in the table as BCVA at the last visit.

In line 232 please make a note that ~75% of cases were misdiagnosed as viral keratitis. Raising light into suspecting amoeba or other pathogens at time of first clinical presentation.

We appreciate this input. We have added that approximately 75% of cases were initially misdiagnosed as viral keratitis.

It would also be good to know if the patients were prescribed steroids before a true clinical diagnosis was made and how many this occurred to. This can exacerbate the infection and could lead to worse clinical outcomes.

Information regarding topical steroid treatment prior to diagnosis has now been included in table 1.

Minor edits

Discrepancy between line 159 “Genotyping was attempted on 22 samples and was successful in 17 samples” and lines 192-193 “Twenty-one samples were available for this study, and DNA was successfully extracted and sequenced in 16 of them.” Where did this other sample come from? Please update records and statistics with including the missing “PHAS11-2 KLAR” sample.

The correct number of culture-positive samples included in the study was 16. One sample, taken from contact lens solution, was excluded because it was considered contamination; the patient was diagnosed with Pseudomonas keratitis and showed prompt improvement with antibiotics only. This is now clarified in the results section.

Please add reference to the figures in the main text and discuss about the level of sequencing and why there are two phylogenetic trees formed.

There is now clarification with reference to the figures in the main text.

Are you able to associate subtype and add these details into Table 2 addendum? Rather than readers trying to extrapolate this themselves from your trees.

The subtypes are now added in Table 2 addendum

Remove “figure E2” from bottom right of Fig 1.

”Figure E2” is now removed.

Line 240 – please add concentrations of these therapies. – 0.08% of PHMB is now being adopted as the standard care based on the ODAK Phase III clinical trial in EU. SIFI will donate PHMB in EU for the treatment of AK.

We are grateful for the observation and have added the concentrations of respective therapies.

Table 2 – You don't need to have the authors written out twice. Look here;

<https://journals.asm.org/journal/spectrum/article-types#research-articles>

“In the text, references are cited parenthetically by number in sequential order”. You can then use the respected number instead of writing out the authors twice. References are different font than main text.

The reference list has now been updated in accordance with the ASM style.

In table 2 addendum, you could add a check box for full or partial 18S sequence and have a checkbox for which samples are then included in which figure.

We have added the suggested check boxes in Table 2 addendum.

Re: Spectrum03058-25R1 (**Acanthamoeba keratitis cases in Sweden: genotypes and clinical course**)

Dear Dr. Elin M Karlsson:

Thank you for the privilege of reviewing your work. Below you will find my comments, instructions from the Spectrum editorial office, and the reviewer comments.

Thank you for addressing many of the issues raised by the reviewers. Please consider these additional points:

1. Reviewer 2's first minor edit was not fully addressed. There is still an inconsistency between line 150, where it indicates there were 22 samples available, and line 184, where it indicates there were 21.
2. Your edit in response to Reviewer 2's second minor edit has introduced another discrepancy. The text added on lines 192-194 indicates only 6 samples were successfully sequenced using the 1375 bp region, and all were successful on the 257 bp region. This is in conflict with the methods, highlighted by the statement on lines 131-132, which indicates that the larger region was done because the smaller region was unsuccessful. Please clarify.
3. Table 2 is not mentioned in the results and does not appear necessary, given that the references are available. Please remove and highlight any other salient details from these studies in the text.
4. Please add a Data Availability paragraph in accordance with ASM policy (<https://journals.asm.org/open-data-policy>) and Reviewer 1's comments.

Revision Guidelines

Sincerely,
Gillian Tarr
Editor
Microbiology Spectrum

Response to Reviewers

This important article raises awareness of Acanthamoeba keratitis cases in Sweden between 2006- 2022. It highlights an important observation with T4 and T3 genotypes being high on the number of AK cases seen within European countries.

Comments

Please remove the second title in line 26.

The second title on line 26 has been removed.

I think it would be best to include the Addendum Figures in the main text of the results of this paper.

The addendum figures have now been included in the main text.

Are these specimens biobanked? Can someone gain access to these cultured strains if they reach out? Can you start biobanking these?

The specimens are biobanked and it is possible to get access to them through the Public Health Agency of Sweden upon request (info@folkhalsomyndigheten.se).

At our Free-living amoeba meeting last month. There was a case of bilateral AK from two different Acanthamoeba species and genotypes. It may be best to try and get samples of both eyes in the future as we may assume it is just one, but this could complicate things significantly for the patient.

We agree that sampling from both eyes is mandatory when bilateral Acanthamoeba keratitis is suspected. In this retrospective study, however, we were limited by the information available in the medical records, and in this particular case only one eye had been sampled.

Major edits

You mention several times that 16-culture positive and then “Genotyping was attempted on 22 samples and was successful in 17 samples”. You include this in Fig 1 Addendum as sample “PHAS11-2 KLAR” but fail to include this in Table 2 or throughout the text.

Thank you for your thorough review. The correct number of culture-positive samples included in the study was 16. One sample was from contact lens solution and was excluded because it was considered a contamination; the patient was diagnosed with Pseudomonas keratitis and showed prompt improvement with antibiotics only. This is now clarified in the results section.

It would be ideal to include the information about the isolate and update the numbers as this seems to be another T4 isolate.

Information about the excluded isolate is now described more clearly in the results section. However, because this isolate was not included in the study, the number of included cases remains 16.

It would be good to include treatment (drugs and %’s administered) and treatment timeline before the surgery column in Table 1.

Thank you for this valuable suggestion. However, since multiple medication combinations were used and treatment regimens were frequently adjusted, reporting the exact sequence and proportions of the administered drugs was not feasible. Therefore, we summarised the medical treatment based on the most commonly used combinations, as described in the Clinical Results section. The concentrations of the most commonly used drops, chlorhexidine (0.02%) and polyhexamethylene biguanide (0.02%), are now specified in the text.

You may also want to include clinical outcome – resolved after Rx, resolved after surgery, enucleated, lost to follow up.

Thank you for the suggestion. The requested clinical outcome information is found in Table 1, which details the type of surgery performed for each patient, including those who underwent evisceration or enucleation. Clinical outcome is also shown in the table as BCVA at the last visit.

In line 232 please make a note that ~75% of cases were misdiagnosed as viral keratitis. Raising light into suspecting amoeba or other pathogens at time of first clinical presentation.

We appreciate this input. We have added that approximately 75% of cases were initially misdiagnosed as viral keratitis.

It would also be good to know if the patients were prescribed steroids before a true clinical diagnosis was made and how many this occurred to. This can exacerbate the infection and could lead to worse clinical outcomes.

Information regarding topical steroid treatment prior to diagnosis has now been included in table 1.

Minor edits

Discrepancy between line 159 “Genotyping was attempted on 22 samples and was successful in 17 samples” and lines 192-193 “Twenty-one samples were available for this study, and DNA was successfully extracted and sequenced in 16 of them.” Where did this other sample come from? Please update records and statistics with including the missing “PHAS11-2 KLAR” sample.

The correct number of culture-positive samples included in the study was 16. One sample, taken from contact lens solution, was excluded because it was considered contamination; the patient was diagnosed with Pseudomonas keratitis and showed prompt improvement with antibiotics only. This is now clarified in the results section.

Please add reference to the figures in the main text and discuss about the level of sequencing and why there are two phylogenetic trees formed.

There is now clarification with reference to the figures in the main text.

Are you able to associate subtype and add these details into Table 2 addendum? Rather than readers trying to extrapolate this themselves from your trees.

The subtypes are now added in Table 2 addendum

Remove “figure E2” from bottom right of Fig 1.

”Figure E2” is now removed.

Line 240 – please add concentrations of these therapies. – 0.08% of PHMB is now being adopted as the standard care based on the ODAK Phase III clinical trial in EU. SIFI will donate PHMB in EU for the treatment of AK.

We are grateful for the observation and have added the concentrations of respective therapies.

Table 2 – You don't need to have the authors written out twice. Look here;

<https://journals.asm.org/journal/spectrum/article-types#research-articles>

“In the text, references are cited parenthetically by number in sequential order”. You can then use the respected number instead of writing out the authors twice. References are different font than main text.

The reference list has now been updated in accordance with the ASM style.

In table 2 addendum, you could add a check box for full or partial 18S sequence and have a checkbox for which samples are then included in which figure.

We have added the suggested check boxes in Table 2 addendum.

Respos to reviewers, part 2

1. Reviewer 2's first minor edit was not fully addressed. There is still an inconsistency between line 150, where it indicates there were 22 samples available, and line 184, where it indicates there were 21.

Thank you for pointing this out. It was a typographical error that we overlooked. The correct number of samples is 21. We apologise for the oversight. This has now been corrected.

2. Your edit in response to Reviewer 2's second minor edit has introduced another discrepancy. The text added on lines 192-194 indicates only 6 samples were successfully sequenced using the 1375 bp region, and all were successful on the 257 bp region. This is in conflict with the methods, highlighted by the statement on lines 131-132, which indicates that the larger region was done because the smaller region was unsuccessful. Please clarify.

Thank you for pointing out this inconsistency. The discrepancy resulted from an error during revision. To clarify, two additional PCR reactions were preformed targeting the larger 1375 bp and 1750 bp region for samples in which sequencing of the smaller JDP region was possible but of suboptimal quality. The text has now been corrected and clarified in the method section only.

3. Table 2 is not mentioned in the results and does not appear necessary, given that the references are available. Please remove and highlight any other salient details from these studies in the text.

Thank you for this comment. We have updated the text to explicitly refer to Table 2. Although the studies are cited in the references, we believe that Table 2 provides a clear and concise overview that facilitates comparison across studies and improves readability. We therefore consider the table to add value and propose to retain it in the manuscript.

4. Please add a Data Availability paragraph in accordance with ASM policy (<https://journals.asm.org/open-data-policy>) and Reviewer 1's comments.

A Data Availability paragraph has now been added to the Materials and Method section in accordance with ASM policy and Reviewer 1's comments. The specimens analysed in this study are biobanked and available upon reasonable request through the Public Health Agency of Sweden; however, the associated clinical and registry data are not publicly accessible due to ethical and legal restrictions.

In addition to the revisions made in response to the reviewers' comments, a few minor linguistic adjustments have been made to the manuscript.

Re: Spectrum03058-25R2 (**Acanthamoeba keratitis cases in Sweden: genotypes and clinical course**)

Dear Dr. Elin M Karlsson:

Thank you for addressing all comments.

Your manuscript has been accepted, and I am forwarding it to the ASM production staff for publication. Your paper will first be checked to make sure all elements meet the technical requirements. ASM staff will contact you if anything needs to be revised before copyediting and production can begin. Otherwise, you will be notified when your proofs are ready to be viewed.

Sincerely,
Gillian Tarr
Editor
Microbiology Spectrum